# Superfluid response of two-dimensional filamentary superconductors

Giulia Venditti[1][†][*], Ilaria Maccari[2][†], Alexis Jouan[3], Gyanendra Singh[3,4], Ramesh C. Budhani[5], Cheryl Feuillet-Palma[3], Jérôme Lesueur[3], Nicolas Bergeal[3], Sergio Caprara[6,7][*], and Marco Grilli[6,7][*]

**1** SPIN-CNR Institute for Superconducting and other Innovative Materials and Devices, Area della Ricerca di Tor Vergata, Via del Fosso del Cavaliere 100, 00133 Rome, Italy
**2** Department of Physics, Stockholm University, Stockholm SE-10691, Sweden
**3** Laboratoire de Physique et d'Étude des Matériaux, ESPCI Paris, PSL University, CNRS, Sorbonne Université, Paris, France
**4** Institut de Ciéncia de Materials de Barcelona (ICMAB-CSIC), Campus de la UAB, 08193 Bellaterra, Catalonia, Spain
**5** Department of Physics, Morgan State University, Baltimore, Maryland 21210, USA
**6** Dipartimento di Fisica, Università di Roma "Sapienza", P.le Aldo Moro 5, I-00185 Roma, Italy
**7** CNR-ISC, via dei Taurini 19, I-00185 Roma, Italy
[†] These authors contributed equally.
[*] giulia.venditti@spin.cnr.it   marco.grilli@roma1.infn.it   sergio.caprara@roma1.infn.it

April 17, 2023

## Abstract

**Different classes of low-dimensional superconducting systems exhibit an inhomogeneous filamentary superconducting condensate whose macroscopic coherence still needs to be fully investigated and understood. Here we present a thorough analysis of the superfluid response of a prototypical filamentary superconductor embedded in a two-dimensional metallic matrix. By mapping the system into an exactly solvable random impedance network, we show how the dissipative (reactive) response of the system non-trivially depends on both the macroscopic and microscopic characteristics of the metallic (superconducting) fraction. We compare our calculations with resonant-microwave transport measurements performed on $LaAlO_3$/$SrTiO_3$ heterostructures over an extended range of temperatures and carrier densities finding that the filamentary character of superconductivity accounts for unusual peculiar features of the experimental data.**

# 1   Introduction

The availability of low-dimensional compounds exhibiting superconductivity is steadily increasing, often displaying unconventional behaviours in their physical properties. The unavoidable presence of (even weak) microscopic disorder in the vast majority of two-dimensional (2D) materials, as well as other external and/or internal electronic interactions, can fragment the SC condensate leading to inhomogeneity on a mesoscopic scale. In particular, there is increasing evidence that in several classes of low-dimensional superconducting (SC) systems the strongly inhomogeneous nature of the electronic condensate appears as a filamentary SC pattern. Inhomogeneous superconductivity can indeed result from several different mechanisms, where the competition of the SC order parameter with other phases can act as a primary source of filamentarity. This is the case for the competition with charge-density waves in high-temperature SC cuprates [1,2], in Cu-intercalated $TiSe_2$ [3,4] and in $HfTe_3$ [5]. Hints of filamentary superconductivity have been observed also in the low-temperature antiferromagnetic phase of Fe-based superconductors [6–11], persisting until the long-range antiferromagnetic order is completely suppressed. The clustering of superconducting electrons into anisotropic stripe-like or puddle-like geometry is found also in $WO_{2.90}$ probably caused by the presence of $W^{5+}$ - $W^{5+}$ electron bipolarons [12]. Alongside chemical doping, also gating fields can trigger phase separation leading to a filamentary SC condensate. For instance, the ion-liquid gating technique, used to inject and tune the number of carriers in systems such as transition metal dichalcogenides (TMD) and transition metal nitride (TMN), can induce a negative compressibility while acting as a primary source of phase separation [13]. One paradigmatic class of materials displaying a strong anisotropy of superconducting regions in their two-dimensional electron gas (2DEG) are $SrTiO_3$-based heterostructures, like, e.g., $LaAlO_3/SrTiO_3$ (LAO/STO). The inhomogeneities in these systems can be ascribed to various causes related to oxygen vacancies [14,15], to the so-called *polar catastrophe* caused by the abrupt polar discontinuity between stacked planes [16,17], to a kind of combination of them [18] or to the sizable Rashba spin-orbit coupling [19,20].

    While the microscopic origin of filamentary superconductivity depends on the specific nature of the system under investigation, some properties of the emergent SC condensate are generically

related to its filamentary inhomogeneous nature, the study of which is therefore of interest to a very broad class of systems. Indeed, it has already been discussed how the anomalous transport properties observed in some inhomogeneous superconductors can be almost entirely ascribed to spatial inhomogeneities of the condensate on a mesoscopic scale rather than to its microscopic nature. That is the case, for instance, of the large broadening of the resistive transition, which cannot be ascribed to paraconductivity effects [21], but is instead the hallmark of the percolating nature of the SC transition. Likewise, the observation on non-linear *IV* characterics [22] or pseudo-gap signatures in the tunneling spectra [23] at temperatures higher than $T_c$ have also been connected with the physics of inhomogeneities.

While several studies have focused on the effect of the mesoscale inhomogeneity on the SC transition above the critical temperature, few studies have investigated the superfluid response of the resulting filamentary condensate [24, 25]. In this paper, we face this issue by mapping the problem into a random-impedance network (RIN) model that we solve exactly. By studying different RIN realizations, we show how the superfluid response of the system non-trivially depends on its microscopic and macroscopic characteristics. At the same time, by comparing our theoretical results with complex conductivity measurements on LaAlO$_3$/SrTiO$_3$ (LAO/STO) interfaces, we show how the different doping regimes can be understood in terms of an intrinsically less or more robust filamentary SC condensate.

The paper is organized as follows. In Section 2, we introduce the problem of the superfluid stiffness in a filamentary superconductor. In Section 3, we discuss the RIN model implemented to study the odd features that can arise from a fractal-like geometry of the superconducting condensate. Sections 4 and 4.1 are devoted to the specific case of LaAlO$_3$/SrTiO$_3$ (LAO/STO) interfaces, summarizing what has been done and what are the unconventional observations of superfluid density and residual conductivity. Finally, in Section 5 we present our theoretical results and in Section 6 our conclusions.

## 2   Filamentary superconductivity

Disregarding the specific microscopic origin of inhomogeneities, we aim at investigating the superfluid stiffness response of a filamentary superconductor.

According to the Bardeen-Cooper-Schrieffer (BCS) theory, in conventional superconductors the energy scale $\Delta$ – associated with the formation of the Cooper pairs – is much smaller than the superfluid stiffness $J_s$ – associated with the global phase rigidity of the SC condensate. Being $\Delta \ll J_s$, the superconducting transition is thus essentially driven by the suppression of $\Delta$. This scenario holds even in the presence of strong disorder and partially inhomogeneous systems [26, 27]. At the same time, in BCS conventional superconductors, $J_s \approx E_F$ ($E_F$ being the Fermi energy of the metal) is directly proportional to the number of superfluid carriers $n_s$, with $J_s \sim n_s/m^*$, and $m^*$ the effective carrier mass. Due to this proportionality, $J_s$ and $n_s$ are often used synonymously. However, it is important to emphasize here that superfluid density and superfluid stiffness are generally two separate quantities. That is clear in those SC systems where, being $J_s < \Delta$, the SC phase transition is driven by phase fluctuations that destroy the phase rigidity of the condensate while preserving a finite density of paired carriers. Granular superconductors and some Josephson-junction arrays are well-known examples of such cases. In 2D superconductors, the leading role of phase fluctuation emerges clearly in the Berezinskii-Kosterlitz-Thouless (BKT) theory [28–30] that provides a clear picture of the SC transition in terms of vortex-antivortex bind-

ing [27]. The BKT fingerprints in real SC systems can be, however, partially or completely masked by the presence of disorder. While spatially-uncorrelated disorder is essentially irrelevant to the BKT SC transition [31, 32], the presence of spatially-correlated inhomogeneities can, indeed, significantly modify its nonuniversal properties [33, 34]. Finally, in some systems, the inhomogeneities are so strong and correlated in space, that the vortex-antivortex unbinding is no longer the leading mechanism for the SC transition [22]. An even stronger role of phase fluctuations takes place in quasi-one-dimensional superconductors, where the so-called phase slips, induced either by thermal or quantum excitations, prevent the onset of a global SC phase coherence [35].

The occurrence of superconductivity on structures made of random nearly 1D filaments, which can intersect and/or go almost parallel, obviously raises the complex issue of phase rigidity both at the local level of single filaments and at the global level of interconnected filaments with more or less pronounced long-range connectivity. While this issue was already addressed some time ago for d.c. transport [36], and in comparing the BKT physics with the effects of inhomogeneities [22], it is of obvious interest to directly investigate, both experimentally and theoretically, the phase rigidity of the condensate in such complex filamentary geometry. In the present work, we precisely aim at studying the superfluid response of a filamentary superconductor, devoting specific attention to the separate role of local and global (geometric) properties and their specific role in determining the complex conductivity response.

Having this goal in mind, we investigate a model system, keeping separate the role of the geometric structure, i.e., the density of filaments and their long-range connectivity, and the role of local disorder, from the role of local superfluid density and conductivity. The former determines the distribution of the local superconducting temperature in the various regions of the system (embedded in an otherwise metallic matrix), while the latter the local stiffness in the single individual pieces of the random superconducting structure.

To keep our study as general as possible, we will assume the filamentary structure to be given from the start, regardless of its microscopic origin.

## 3 Theoretical description: the Random Impedance Network

Several 2D superconducting systems, such as LAO/STO [36, 37], TMD, and TMN [13], exhibit an unusual gradual and broad vanishing of $R(T)$ that cannot be ascribed to conventional SC fluctuations but rather to the emergence of an inhomogeneous SC condensate. The first depletion of $R(T)$ by lowering the temperature can be, indeed, attributed to the appearance of SC puddles, i.e., rather bulky regions that, at lower temperatures, get connected through SC filamentary branches, with long-distance connectivity, ultimately responsible for the long tail of $R(T)$ approaching the critical temperature (see Fig. 1). In what follows, we will refer to the bulky SC puddles with the subscript (b), and to the filamentary SC regions with (f).

In a series of previous works, we demonstrated that d.c. transport in such strongly inhomogeneous compounds can be conveniently described by a Random Resistor Network (RRN) model, in which the 2D system is discretized on a square lattice, where each bond is associated with a resistor. While part of the resistors remains in the normal-metal state down to the lowest attained temperatures, thereby forming a metallic matrix, some others become superconducting when T decreases below their local critical temperature $T_c^{ij}$, where $i$ and $j$ are the neighbouring sites of the square lattice identifying the resistor bond. Specifically, a Gaussian distribution of critical temperatures $T_c^{ij} \neq 0$ was assumed, characterized by a given average value $\mu$ and a variance $\sigma$. An

extended analysis also showed that different statistical distributions provide more or less equivalent physical results [36] and therefore, for the sake of simplicity, we here only consider Gaussian distributions of local critical temperatures. A more precise description of transport data also led to distinguishing between the Gaussian distribution of bulkier (puddle-like) regions, with a mean value of the SC critical temperature ($\mu_b$) slightly higher than the global $T_c$, and a Gaussian distribution of the filamentary regions where the average SC temperature ($\mu_f$) is slightly lower and, in general, more broadly distributed.Once the random inhomogeneous structure of the system has been specified, we define $R_m$ as the resistance of the metallic matrix, while for the superconducting bonds, which include both the filamentary and the puddle-like regions, we assign a resistance value such that $R_{ij}^s(T > T_c^{ij}) = R_s$ and $R_{ij}^s(T < T_c^{ij}) = 0$. While the standard deviations, $\sigma_f$ and $\sigma_b$, relative to the two Gaussian distributions for the filamentary and puddle-like SC regions, determine the extension in temperature of the resistance tail, the spatial filamentary structure of the SC cluster is crucial to recover the $R(T)$ approaching $T_c$ with a long slowly vanishing tail.

So far, the discussion of the consequences of filamentarity on transport has focused only on the metallic phase stressing the 'tailish' behaviour of $R(T)$ as the signature of filamentary SC. In this work, we more directly address the issue of the superconducting response of the system by generalizing the RRN to finite frequencies thereby calculating the complex conductivity of the system. Thus, we assign to each bond a complex impedance $Z_{ij} = R_{ij} + i\omega_0 L_{ij}$, where $\omega_0$ is the frequency of the circuit and $L_{ij}$ is either the inductance of the superconducting bonds, $L_s$, or that of the metallic matrix, $L_m$. The extension to finite frequencies of the RRN model into a Random Impedance Network (RIN) model was already investigated in its effective medium analytical solution [24, 25], where the information on the geometrical structure of the cluster was, however, completely neglected. Here, we calculate exactly the global effective impedance $Z_{tot} = R_{tot} + i\omega_0 L_{tot}$ of the lattice by solving the Kirchhoff and Ohm laws of the network. That allows us to account for the role played by the SC cluster geometry that we generated using a diffusing limited aggregation algorithm, discussed in Appendix A. In order to be quantitative, in the present work we will take as a case study the resonant-microwave measurements performed on LAO/STO interfaces, which we discuss in the next Sections. Our goal is to identify the physical ingredients needed to reproduce the specific peculiarities found in experiments and summarized in the next Section.

# 4 LaAlO$_3$/SrTiO$_3$ interfaces

In SrTiO$_3$-based heterostructures, the carrier density of the 2DEG formed at the interface can be tuned by a gate voltage $V_G$. Despite their very clean and regular structure, these heterostructures have revealed an intrinsic tendency to electronic phase separation [16], leading to the formation of an inhomogeneous superconducting state with a filamentary character [37, 38]. This happens even for the [001] orientation, i.e., when the interface is orthogonal to the $c$-axis of both LAO and STO; henceforth, we will always refer to [001] LAO/STO samples. Tunnelling [23, 39], atomic force microscopy [40], and critical current experiments [41] provide clear evidence of an inhomogeneous superconducting condensate at the LAO/STO interface. Direct measurements of the superfluid density via SQUID measurements [42] showed that $J_s(T)$ does not follow neither BCS nor BKT prescriptions; the behaviour was instead well captured once the inhomogeneous character of the condensate was considered [37]. Transport measurements report further signs of inhomogeneity, with a percolating metal-to-superconductor transition where a sizable fraction of

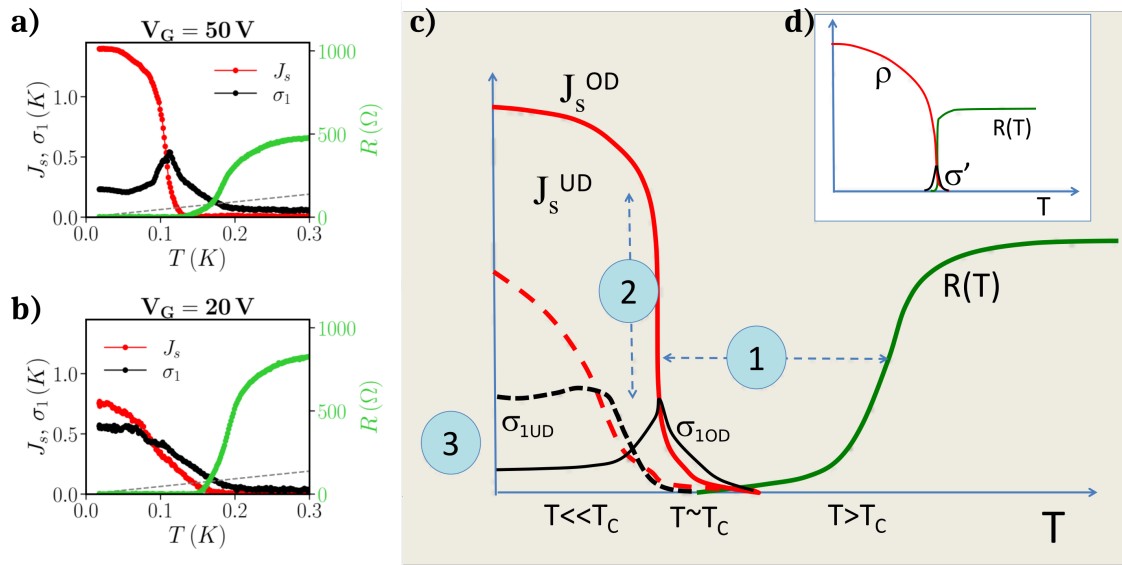

Figure 1: DC resistance (green), superfluid stiffness $J_s \propto \sigma_2$ (red) and optical conductance $\sigma_1$ (black) as functions of temperature. (a) Experimental data for a gate voltage $V_G = 50\,\text{V}$ (OD) and (b) for $V_G = 20\,\text{V}$ (UD). The grey dashed line is the BKT critical line $2T/\pi$. (c) Sketched summary of the observed features in the UD (dashed lines) and OD regimes (solid lines): (1) the broad and tailish transition of $R(T)$ coincide with a very gradual increase of $J_s$ at $T \sim T_c$; this results in a wide separation between the two, paradigmatic of a percolating yet filamentary superconducting cluster; (2) increase of $J_s$ at $T \lesssim T_c$, more abrupt in OD systems than in the UD ones, thus signalling the more or less homogeneous nature of the superconductor at different fillings; (3) the substantial residual value of $\sigma_1$, more important in the UD case yet more peaked in the OD. Those features are clearly at odds with the scenario of a dirty yet homogeneous 2D superconductor, schematically reported in (d).

the 2DEG remains metallic down to the lowest accessible temperature [21,36–38]. The resulting filamentary state, where the SC regions live on 100-nanometer length scales [43], coexist with the linear SC regions identified on the micron scale at structural domain boundaries [44,45]. The length scales at play are in perfect agreement with the effective medium approach used in [37], which accounts for the intrinsic averaging operated by the SQUID device over the micrometric scale.

In a previous publication [46], we reported the results of resonant microwave transport experiments at the lowest temperatures. From a comparison between the gap and the superfluid stiffness energy scales, we identified two distinct regimes: an overdoped (OD) regime (i.e., with a carrier density higher than the one corresponding to the maximum superconducting critical temperature $T_c$), in which the LAO/STO system has the character of a dirty but rather homogeneous two-dimensional (2D) superconductor, and an underdoped (UD) regime (i.e., with a carrier density lower than the one corresponding to the maximum superconducting critical temperature $T_c$), where the superconducting state closely resembled a disordered 2D Josephson-junction array (see Fig. 4 of Ref. [46]). In this work, we consider the resonant microwave transport experiments over a broad temperature and carrier density range. The correspondence between gate voltage and carrier density is given in Fig. 5 of Ref. [46]. By measuring both the complex conductivity, i.e.,

$\sigma = \sigma_1 - i\sigma_2$, and the DC resistivity, we study the superconductor-to-metal transition characterizing the emerging filamentary superconducting state, via the temperature dependence of its superfluid stiffness, $J_s \propto \omega \sigma_2$, and optical conductance, $\sigma_1$.

In Fig. 1, we report the resistance (green) and complex conductivity data (real part in black, imaginary part in red) for a LAO/STO sample. In particular, Fig. 1(a) and 1(b) show two paradigmatic examples for the OD (gate voltage $V_G = 50$ V) and UD ($V_G = 20$ V) regimes, respectively.

In Fig. 1(c), we schematically summarize the peculiar features found in the two doping regimes of LAO/STO samples. Besides the broad transition and tailish behaviour of the resistance curve $R(T)$ (green solid line), near the temperature $T_c$ that marks the transition to a filamentary superconducting state [21, 36–38], we outline three main peculiar features:

(1) going through the metal-to-superconductor transition, an unusual and surprising separation appears between the temperature at which the resistivity vanishes and that relative to the onset of a sizable superfluid stiffness. Both $R(T)$ and $J_s(T)$ show a long tail, symptomatic of a percolating filamentary superconducting state, still too fragile to establish a 2D rigid condensate;

(2) lowering the temperature below $T_c$, the superfluid stiffness shows an increase, steep in the OD case and more gradual in the UD regime; notice that in the OD regime, the steep increase of $J_s$ occurs at a temperature much lower than $T_c$;

(3) at $T \ll T_c$, the real part of the conductance $\sigma_1(\omega_0, T)$ takes a significant residual value. This last feature marks the persistence of a sizable fraction of normal metal down to the lowest temperatures, further supporting the idea of an inhomogeneous superconducting state.

This last feature marks the persistence of a sizable fraction of normal metal down to the lowest temperatures, further supporting the idea of an inhomogeneous superconducting state.

Finally, in Fig. 1(d) we sketch the behaviour of the same quantities as a function of temperature within the BKT scenario in the presence of moderate disorder [47]. The BKT scheme clearly fails to reproduce the main features of the data. Even the seeming jump experimentally observed in the superfluid stiffness, e.g., at $V_G = 50$ V, cannot be interpreted as the paradigmatic hallmark of the BKT transition, rather expected at the intercept with the $2T/\pi$ critical line [dashed-grey line in Fig. 1(a)].

In what follows, we present our extensive resonant microwave transport analysis. We show the actual occurrence of these three peculiar features in the experimental data, and we discuss our theoretical analysis to extract information about the structural characteristics of the superconducting system throughout the temperature and carrier density range considered. It is worth noting that the three peculiar features summarized above are indeed characteristic of the rather disordered sample, whereas in more homogeneous samples some of those peculiarities can be less pronounced or even absent. Nonetheless, although this sample may not be representative of every LAO/STO interface, it offered the motivation to study the effect of filamentarity and to generalize its consequences, without the burden of worrying about microscopic details. Our theoretical investigation provides a coherent rationale for the observed peculiar experimental features in terms of a filamentary superconducting structure embedded in a metallic matrix and following their evolution with carrier density and temperature.

### 4.1 Resonant microwave transport experiment

In this work, we adopted the same sample preparation and experimental setup of Ref. [46] to perform a complex conductivity analysis of a back-gated [001] LAO/STO sample throughout an extensive carrier density and temperature range (experimental details can be found in Appendix B). In Fig. 2b and c we report the real and imaginary part of the complex conductivity as a function of temperature for a resonant microwave frequency $\omega_0/2\pi = 0.36$ GHz and several values of the gate potential. Panel (c) reports the imaginary part of the conductivity $\sigma_2(\omega_0, T)$, proportional to the superfluid stiffness, displaying two markedly different temperature trends, according to the applied gate voltage. In the OD regime, with gating between 28 V and 50 V, the superfluid stiffness grows slowly with reducing the temperature below 0.16 K and then rapidly increases with a downward curvature between $0.11-0.13$ K. In the UD regime, from 26 V down to 8 V, the superfluid stiffness grows much more gradually, with an upward curvature down to lower temperatures $0.04-0.07$ K. A similar dichotomous behaviour is observed in the real part of the conductivity $\sigma_1(\omega_0, T)$ (panel b): in the OD regime, a rather sharp peak is observed at temperatures corresponding to the rapid increase of the superfluid stiffness, while in the UD regime, $\sigma_1(\omega_0, T)$ presents a much broader peak or no peak at all. In both cases, however, the real part of conductivity stays finite at the lowest temperatures, assuming values that are non-monotonic and maximal around gate voltages $\sim 20-24$ V.

This crossover is even more evident if one looks at the whole picture as a function of the gate voltage. While the voltage dependence of the critical temperature $T_c$ vs $V_G$ (green dots in Fig.2a) does not give any clear indications, the temperature at which $\sigma_1$ reaches its maximum value $(T(\sigma_1^{max})$ in grey) gives a rather clear idea of the crossover between the UD to the OD regimes. At the same time, the saturating value of $J_s^{\max} = J_s(T \to 0)$ shows how the system falls outside the theoretical framework of conventional BCS superconductors. Within the standard BCS scenario the superconducting gap at zero temperature $\Delta_0$ is expected to follow the $T_c$ dome, being $\Delta_0 \approx 1.76 k_B T_c$. On the other hand, assuming the dirty limit, the same gap should behave as $\Delta_0 \sim J_s(0)R_N$. In Fig. 2 we display in red (right axis) the quantity $\Delta^{\exp} = 4e^2 J_s R_N/\hbar\pi$, to underline once again the discrepancy of such measurements with the BCS scenario [46]. The deviation observed is consistent with our idea of filamentary superconductivity presented in Section 2.

## 5 Theoretical results and their interpretation

Despite its conceptual simplicity, the RIN model is complete and flexible enough to reproduce the rather unconventional trends experimentally observed. By lowering $T$, the bonds with $T_c^{ij} \geq T$ become superconducting, so that the SC network nucleates inside the normal-metal matrix with specific signatures depending on the geometric structure (more or less dense filaments), the disorder (represented by the width of the random distribution of $T_c^{ij}$), and on the characteristics of the mesoscopic metallic/superconducting regions, as modelled by the parameters $R_m, L_m, R_s, L_s$. We anticipate that the choice of the values for the "microscopic" resistors and impedances are made to match the experimental measurements at our disposal assuming an angular frequency $\omega_0 = 2 \times 10^9 \, \mathrm{s}^{-1}$. More details can be found in Appendix C. Our goal is to identify which specific feature of the model is responsible for each peculiar feature of the real system.

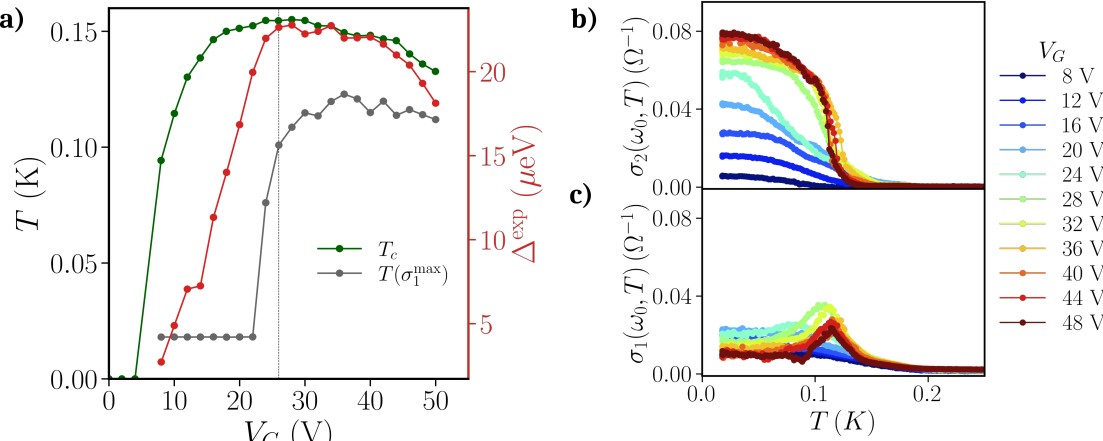

Figure 2: (a) Typical dome $T_c$ vs $V_G$ (green dots) observed in LAO/STO interfaces compared to the temperature of the peak in $\sigma_1$ in Kelvin ($T(\sigma_1^{\mathrm{max}}$ grey dots). In red (right axis) we display $\Delta^{\mathrm{exp}} = 4e^2 J_s R_N / \hbar \pi$. In a bare BCS scenario, this should be proportional to the dome. The dashed vertical line indicates the crossover from the UD and the OD system at $V_G = 26$ V. (b) Imaginary $\sigma_2(\omega_0, T)$ and (c) real part $\sigma_1(\omega_0, T)$ of the complex conductivity as functions of temperature and at various gate voltages. The microwave frequency is $\omega_0 / 2\pi = 0.36$ GHz.

## 5.1  Effect of the geometry and disorder on the resistance and superfluid behaviours

The geometry of the superconducting cluster and the widths $\sigma_b$ and $\sigma_f$ of the random distribution, $P(T_c^{ij})$, of the individual SC bonds critical temperatures $T_c^{ij}$ encode the most prominent peculiar property (1) of the LAO/STO superconductor [Fig. 1(c)]. Starting from the normal state, by lowering the temperature, the resistance smoothly decreases, mostly due to the puddle-like regions becoming superconducting; if these were absent, with a SC cluster only made of filaments, the decrease of $R(T)$ would indeed start in a much sharper way. We thus investigated the relevance of both filaments and puddles.

The filamentary structure is built via a diffusion-limited aggregation (DLA) algorithm [13, 36] (see Methods for details). We stress here that extensive iterations of the DLA algorithm would produce a fractal-like geometrical structure, but in our case this is instead a mere technical tool to produce a random assembly of filamentary structures on our finite square-lattice cluster. The superconducting puddles, with a given radius $r_{pd}$, are afterwards added to the cluster, to reach the total superconducting density $w$ we fixed. It follows that the larger $r_{pd}$, the less numerous the puddles will be. Their role is fundamental in explaining the first downturn of $R(T)$ but, once they became superconducting, their size is almost irrelevant to the complex conductivity properties. Indeed, the superfluid rigidity and the residual dissipation are mainly determined by the structure and the density of the filamentary components of the superconducting cluster, while the puddles play a minor role. Specifically, by lowering $T$, the filamentary structures become more and more superconducting and, when a superconducting percolating path forms, the resistance vanishes at the global critical temperature $T_c$, $R(T_c) = 0$. Due to the nearly one-dimensional (1D) character of the filamentary structures and their poor connectivity, the resistance stays low but finite until the very last resistor of the percolating path is switched off. That explains why the filamentary geometry is crucial to account for the tailish behaviour of $R(T)$.

How broad the transition and how long the tail depends instead on the width $\sigma_{f,b}$ of the Gaussian distribution of the $T_c^{ij}$s, accounting for the microscopic impurities generically present in real systems. At the same time, the filamentary percolating cluster is formed by a low fraction of superconducting bonds with a nearly 1D structure. Therefore, they cannot result in a large rigidity of the superconducting condensate. Indeed, a rather small $\sigma_2 \propto J_s$ is found for $T \lesssim T_c$. The long tails observed in LAO/STO interfaces in both $R(T)$ and $J_s(T)$, which also lead to the apparent separation of the two curves, find in this way a natural explanation [Fig. 1(c)].

By further lowering the temperature, more and more bonds in the random filamentary subset become superconducting, leading to the more or less rapid growth of the condensate rigidity depending on the specific features of the superconducting subset. Quite obviously, the more or less dense (and interconnected) character of the filamentary structure determines the more or less rapid growth and the intensity of the condensate rigidity $J_s$ (See Appendix A).

## 5.2 Effects of the internal character of the mesoscopic metallic and superconducting regions

Besides the effect of the geometry and density of the filaments, the more or less rapid growth of $\sigma_2 \propto J_s$ at some $T < T_c$ – peculiarity (2) of Fig. 1(c) – is also dependent on the internal rigidity of the individual mesoscopic superconducting bonds, via their parameter $L_s$. The smaller the local inductance $L_s$, the more rigid the individual mesoscopic superconducting bond and the more rapidly and intensely the overall rigidity grows.

In Fig. 3, we show how both the two experimentally observed regimes, OD and UD, can be successfully captured by fixing the geometric structure of the superconducting cluster, shown in Figs. 3(b) and 3(e), whose filamentary character keep $R(T)$ and $J_s(T)$ well separated. By simply varying the value of the inductances $L_s$ and the width of the random distribution $P(T_c^{i,j})$ for both the filamentary, $\sigma_f$, and the puddle-like SC regions, $\sigma_b$. For the OD regime [see Fig. 3(a)], by fixing $L_s = 0.7$ nH, $\sigma_f = 0.02$ K, and $\sigma_b = 0.03$ K, we recover both the steep increase of $\sigma_2(T)$ (red curve) and the peak of the optical conductivity $\sigma_1(T)$ (black curve) found experimentally. At the same time, for the UD regime [see Fig. 3(d)], we recover the slow increase of $\sigma_2(T)$ as well as the much broader peak of $\sigma_1(T)$ by employing a larger value of $L_s = 2.0$ nH and ima slightly wider distribution of critical temperatures for the filamentary SC bonds, with $\sigma_f = 0.05$ K [see Figs. 3(f) and 3(c)].By comparing our calculations with the experimental results, we can affirm that: a) disorder, i.e., the width of the $T_c^{ij}$ distributions, is comparatively larger in UD systems and b) the local mesoscopic superconducting regions, in the UD regime, have a smaller intrinsic rigidity, i.e., a larger inductance, likely as a consequence of a lower carrier density.

Finally, we address the issue of the substantial residual normal-state real conductivity at $T \ll T_c$ [peculiarity (3) of Fig. 1(c)]. According to our analysis (see also Appendix A and C), we found that at low temperatures the resistance of the residual metallic bonds largely determines the real (dissipative) part of the complex conductivity, with the residual $\sigma_1$ being inversely proportional to $R_m$. At the same time, a sparser geometry of the filaments will result in a more abundant residual metallic component, hence enhancing the dissipation in the superconducting state. The use of different values for the internal character of the resistors, $R_s$ and $R_m$, reflects the presence in [001] LAO/STO samples of two different types of carriers, with different mobilities, whose relative density depends on the gating applied (see Appendix C for details).

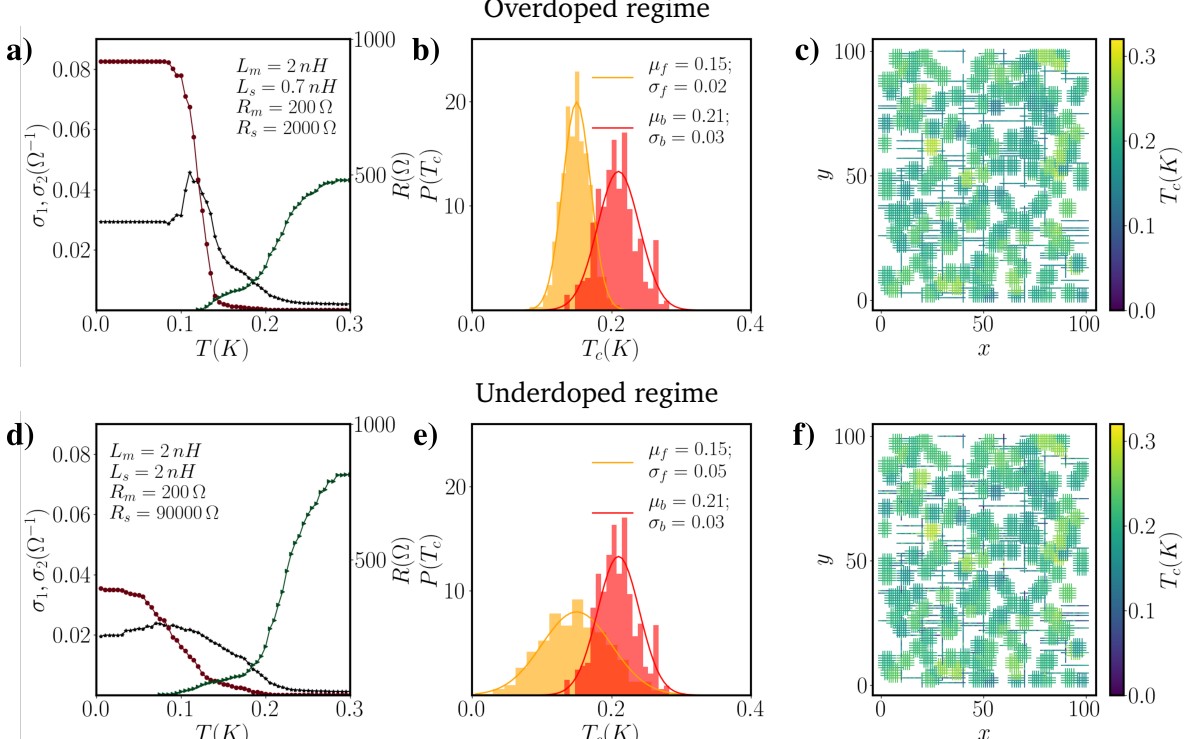

Figure 3: Temperature dependence of complex conductivity and DC resistivity calculated with the RIN model to describe the (a) OD and (d) UD system (real part in black, imaginary part in red and DC resistivity in green). The superconducting structure to which they correspond are shown in panels (c) and (f) respectively; the colour code refers to the local critical temperatures, yellow to blue regions are superconducting, while the metallic matrix is in the white background. Both cases in (a) and (d) refer to the same geometry of the underlying RIN, with total SC density $w = 0.43$, and the same parameters of the metallic matrix $R_m = 200\,\Omega$, $L_m = 2\,\text{nH}$. Instead, the parameters of the superconducting cluster are different: (a) OD: $R_s = 2000\,\Omega$ $L_s = 0.7\,\text{nH}$ (d) UD: $R_s = 90000\,\Omega$ $L_s = 2\,\text{nH}$ as well as the width of the $T_c^{ij}$ distribution for the filamentary SC regions, as visible from the corresponding panels in which (b) OD: $\sigma_b = 0.03\,\text{K}$, and $\sigma_f = 0.02\,\text{K}$ and (e) UD: $\sigma_b = 0.03\,\text{K}$, and $\sigma_f = 0.05\,\text{K}$. This last difference is highlighted in panels (c) and (f) where we show the corresponding distributions of $T_c^{ij}$ for the puddles and the filamentary structure.

# 6   Discussion and concluding remarks

In summary, we presented here a detailed theoretical interpretation of the complex conductivity experimentally measured in a [001] LAO/STO interface. Our theoretical analysis shed light on the intriguing peculiar features experimentally observed, revealing that they stem from the interplay between the filamentary structure of the superconducting cluster, embedded in a normal metal, and disorder, resulting in a random distribution of local critical temperatures. The main consequence is that the superfluid properties, in particular, the rigidity of the condensate, primarily depend on the geometrical structure of the superconducting cluster and only secondarily on the density of the superfluid matter. This result is highly nontrivial since the concepts of superfluid density and stiffness are often used as synonymous. We point out that, by neglecting the role of phase fluctuations, that reduce the superfluid stiffness without affecting the density of carriers, this identity is only true for homogeneous systems and can be strongly violated when the system is highly inhomogeneous. Our LAO/STO interface can therefore be taken as an example for a new paradigm of superconducting matter. We are aware that other interfaces and low-dimensional superconductors do not always display the same peculiar features, but we here point out precisely the physical ingredients leading to such anomalous behaviours, which may or may not be present depending on the amount of inhomogeneous charge distribution (resulting in regions with different local critical temperatures) and its more or less filamentary spatial structure.

The very starting point of the model, where disorder is encoded both in the randomly generated filamentary-puddle superconducting cluster and in a random distribution of local critical temperatures, may seem at odds with the very clean and structurally ordered LAO/STO interface. However, the presence of filamentary superconducting regions in [001] LAO/STO interfaces has been experimentally assessed both at the micron [44,45] and at the submicron [43] scales and it is supported by many experimental and theoretical evidence.

The main message of this work is that the peculiar features of the complex conductivity data arise from the inhomogeneous filamentary character of the superconducting regions and the main differences between OD and UD systems stem both from the more or less broad distribution of local $T_c$'s (i.e from the relative relevance of disorder) and from the microscopic characteristics resulting in different values of the parameters $L_s, L_m, R_s, R_m$. In particular, we were able to identify the specific physical effects of each handle of the model on macroscopic transport: the resistivity and inductance of the various regions, how rapidly the normal metal becomes superconducting by decreasing $T$, due to the width of the $T_c^{i,j}$ distribution associated with the microscopic disorder, and so on.

Finally, beyond its theoretical understanding, the study of inhomogeneous filamentary electronic condensates can pave the way for a systematic control and exploitation of superfluid systems with extremely small phase rigidity. This may result in interesting applications for sensors; systems with stiffness that can be tuned by gating and/or temperature; or where the features of a Josephson-junction array can continuously be tuned from nearly homogeneous BCS to quasi-1D superconductors. Last but not least, filamentary superconductors in the presence of large Rashba spin-orbit coupling could provide a new path for the emergence and observation of Majorana fermions [48].

# Acknowledgements

**Author contributions**   G.S. and A.J. performed the measurements assisted by N.B. Samples were fabricated by P.K. and E.L. under the supervision of A.D., R.C.B., A.B., and M.B., G.S., A.J., and N.B. carried out the analysis of the results. S.C. and M.G. elaborated on the model and the generalization of the RRN to finite frequencies. G.V. and I.M. performed the theoretical calculations. The manuscript was written by S.C., M.G., G.V., I.M., and N.B. with contributions and suggestions from all coauthors.

**Funding information**   S.C. and M.G. acknowledge financial support from the Italian Ministero dell'Università e della Ricerca, through the Project No. PRIN 2017Z8TS5B, the PNRR project 'Topological Phases of Matter, Superconductivity, and Heterostructures' Partenariato Esteso 4 - Spoke 5 (n. PE4221852A63A88D), and from the 'University Research Projects' of the Sapienza University of Rome: 'Superconductivity, inhomogeneity and novel phenomena in two-dimensional materials' (n. RM11916B56802AFE), 'Equilibrium and out-of-equilibrium properties of low-dimensional disordered and inhomogeneous superconductors' (n. RM12017 2A8CC7CC7), 'Competing phases and non-equilibrium phenomena in low-dimensional systems with microscopic disorder and nanoscale inhomogeneities' (n. RM12117A4A7FD11B), 'Models and theories from anomalous diffusion to strange-metal behavior' (n. RM12218162CF 9D05). I.M. acknowledges the Carl Trygger foundation through grant number CTS 20:75. N.B. acknowledges the ANR QUANTOP Project-ANR-19-CE470006 grant.

# A   Geometry of the superconducting network

The generation of the superconducting filamentary structure is obtained by means of a diffusion-limited aggregation (DLA) algorithm [13, 36]. Of course, this choice is arbitrary and it does not rest on a straight physical reason nor it aims at demonstrating that the superconducting regions have some defined fractal-like structure. It is simply a technical way to represent strongly inhomogeneous systems with space correlation and connectivity over large distances. The fractal-like structure is grown by diffusing from left to right $n_{RW}$ random walk particles in a square of size $L_{\square}$ larger than the size ($L \times L$) of the square lattice network ($L_{\square} > L$), that we investigate in the complex conductivity calculations. We allow each of the $n_{RW}$ to move $r_{DLA}$ bonds (steps) to the right and $y_{DLA}$ bonds up or down, with equal probability, whereas in the RRN calculations presented in Refs. [13, 36] $r_{DLA} = y_{DLA} = 1$. Hence, we can construct a more or less dense network of filaments just by tuning those parameters, keeping a higher fraction of the metallic residue without preventing percolation.

This procedure is iterated until the particle stops, as soon as it reaches the top, bottom or right edge where it sticks; if it reaches a site already occupied by one of the previously diffused particles, it takes a step back and stops thereby increasing the cluster of aggregated particles: the cluster obtained is defined by all the bonds connecting two stuck particles. From this super-network, a sub-network of size $100 \times 100$ is selected and it will be the superconducting backbone of the RIN. Then, patches of radius $r_{pd}$ will be superimposed until a fraction $w$ of superconducting resistors is reached. In Fig. 4, are shown two $250 \times 250$ different super-network constructed launching $n_{RW} = 15000$ particles. In panel a, the (orange) superconducting fractal is built from random walkers allowed to do $r_{DLA} = 10$ steps on the right and $y_{DLA} = 10$ steps on the left, the same used

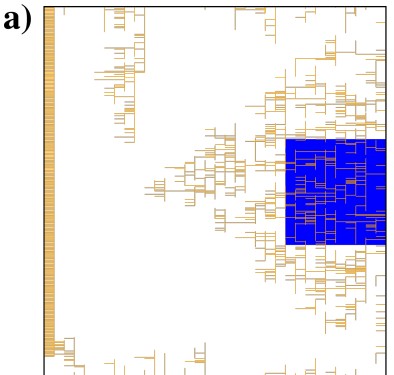
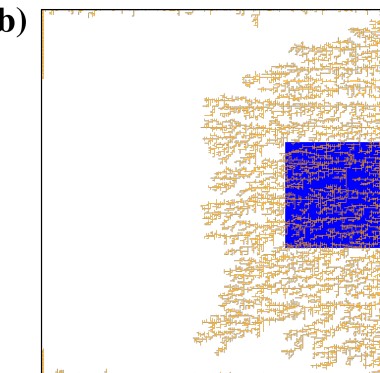

Figure 4: Examples of filamentary structures constructed via the "improved" DLA algorithm launching $n_{rw}$ =15 000 diffusing particles across a $350 \times 350$ square lattice. In orange are shown the obtained clusters with (a)$r_{DLA} = 10$, $y_{DLA} = 10$ and (b) $r_{DLA} = 2$, $y_{DLA} = 2$. Highlighted in blue is the metallic region that defines the final $100 \times 100$ square lattice.

for the results shown in Fig.3, while in panel b the constraints were $r_{DLA} = 2$, $y_{DLA} = 2$. In both panels, the region coloured in blue is the metallic background of the final $100 \times 100$ network.

For the sake of completeness, we show here how a denser fractal geometry modifies the superfluid and resistive responses. In Fig. 5 we present the RIN results obtained for a cluster constructed from a $r_{DLA} = 2$, $y_{DLA} = 2$ fractal, all other parameters being equal to the ones used in Fig. 3. By looking at panels a and d of Fig. 5 one can observe how the shapes of the curves $\sigma_1$, $\sigma_2$ as functions of the temperature are qualitatively different from their counterparts presented in Fig. 3. In particular, the saturating value of $\sigma_2$ is increased by the denser geometry of the underlying fractal. Concerning instead the optical conductivity $\sigma_1$, one can observe that the saturating value at $T = 0$ is unchanged with respect to the geometry of the fractal, whereas its generic behaviour and, particularly, its peak are non-trivially dependent on the filamentary geometry. That occurs despite the fact that the total number of superconducting bonds is the same in all four cases presented in Figs. 3 and 5, being $w = 0.43$, revealing once again the fallacy, in inhomogeneous systems, of the assumption that superfluid density is equivalent to superfluid stiffness. One can also note that the probability distributions $P(T_c)$ (panels b and e) are only slightly changed by the different geometry.

## B    Resonant microwave transport experiment

In this study, we used 8-uc-thick LaAlO$_3$ epitaxial layers grown on $3 \times 3$ mm 2 TiO$_2$-terminated (001) SrTiO$_3$ single crystals by pulsed laser deposition. The substrates were treated with buffered hydrofluoric acid to expose TiO$_2$-terminated surface. Before deposition, the substrate was heated to 830 °C for 1 h in an oxygen pressure of $7.4 \times 10^{-2}$ mbar. The thin film was deposited at 800 °C in an oxygen partial pressure of $1 \times 10^{-4}$ mbar. The LaAlO$_3$ target was ablated with a KrF excimer laser at a rate of 1 Hz with an energy density of 0.56–0.65 Jcm$^{-2}$. The film growth mode and thickness were monitored using reflection high-energy electron diffraction (STAIB, 35 keV) during deposition. After the growth, a weakly conducting metallic back-gate of resistance ~100 kΩ (to avoid microwave shortcut of the 2DEG) is deposited on the backside of the 200 $\mu$m-thick SrTiO$_3$

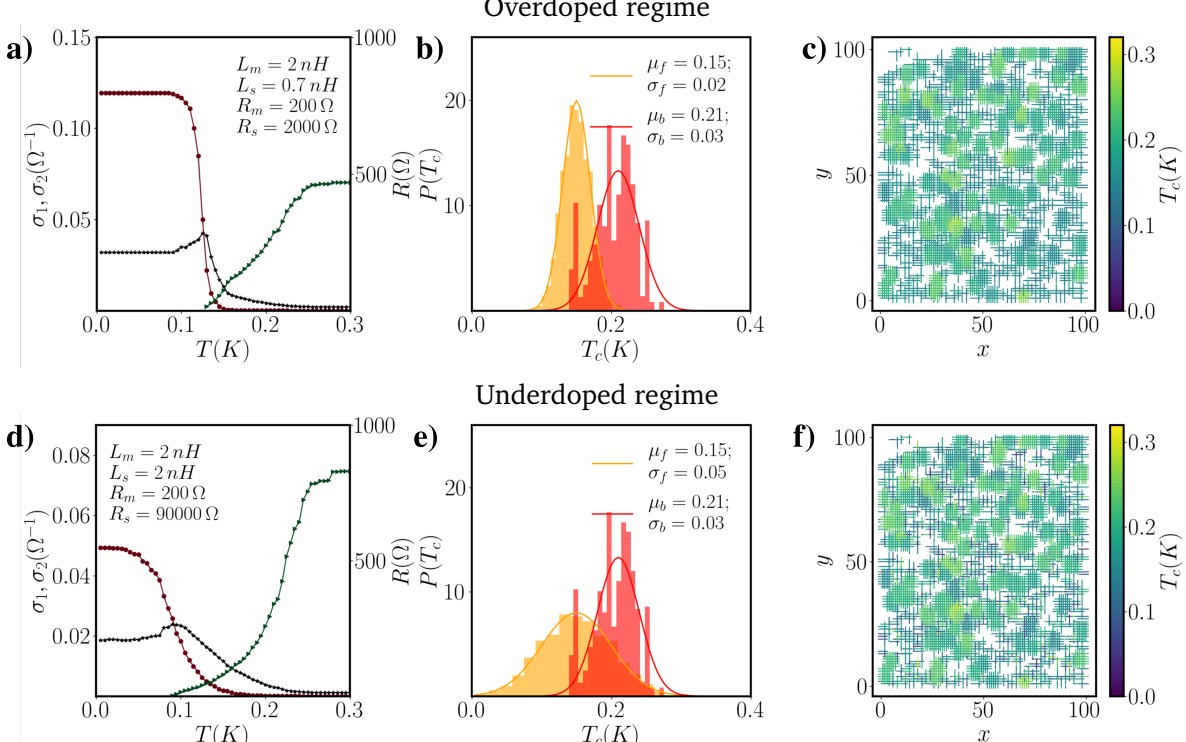

Figure 5: Temperature dependence of complex conductivity and DC resistivity calculated with the RIN model using the same parameters and probability distributions of Fig. 3 but with a denser fractal geometry. (a) Same parameters used to describe the OD regime and (d) the UD regime (real part in black, imaginary part in red and DC resistivity in green). The superconducting structure to which they correspond are shown in panels (c) and (f) respectively; the colour code refers to the local critical temperatures, yellow to blue regions are superconducting, while the metallic matrix is in the white background. Both cases in (a) and (d) refer to the same geometry of the underlying RIN, with total SC density $w = 0.43$, and the same parameters of the metallic matrix $R_m = 200\,\Omega$, $L_m = 2\,$nH. Instead, the parameters of the superconducting cluster are different: (a) OD: $R_s = 2000\,\Omega$ $L_s = 0.7\,$nH (d) UD: $R_s = 90000\,\Omega$ $L_s = 2\,$nH as well as the width of the $T_c^{ij}$ distribution for the filamentary SC regions, as visible from the corresponding panels in which (b) OD: $\sigma_b = 0.03\,$K, and $\sigma_f = 0.02\,$K and (e) UD: $\sigma_b = 0.03\,$K, and $\sigma_f = 0.05\,$K. This last difference is highlighted in panels (c) and (f) where we show the corresponding distributions of $T_c^{ij}$ for the puddles and the filamentary structure.

substrate. The sample was then inserted into a parallel RLC electrical resonant circuit made with surface mount devices (SMD) to perform microwave measurement in a reflection configuration as already used to probe the $SrTiO_3$ and $KTaO_3$ based superconducting interfaces [46, 49, 50]. A bias-tee allows measuring both the DC and AC microwave transport properties of the 2DEG at the same time. After a calibration procedure, the complex conductivity $\sigma = \sigma_1 - i\sigma_2$ of the 2DEG is extracted from the frequency and the width of the resonant peak, which are controlled by the total inductance and the total resistance of the circuit, respectively. The temperature dependence of $\sigma_1$ and $\sigma_2$ for different back gate voltages are reported in Fig. 6.

## C   Choice of the parameters

Besides its overall geometrical structure – filamentary density and broadness of the $T_c^{ij}$ distribution – the model is endowed with local parameters characterizing the transport properties of the individual mesoscopic regions, both the metallic $(R_m, L_m)$ and the superconducting $(R_s, L_s)$ ones. While $L_m$ plays a minor role at any temperature, the resistivity of the metallic regions is crucial to determine the dissipative residual character of the system at low temperature: the lower is $R_m$, the higher is $\sigma_1$ [peculiarity (3) in Fig. 1(d)]. At the same time, the value of $R_s$ is immaterial in the same low-$T$ regime, but is fundamental in fitting the resistivity in the overall normal state $R(T > T_c)$. $L_s$, instead, determines the local rigidity of the condensate inside each mesoscopic superconducting region and plays a relevant role in determining the global rigidity: the lower is $L_s$ the higher is the saturating value of $\sigma_2$ and the steeper is its growth. The choice of $R_s$ and $R_m$ becomes rather stringent in UD systems, where the large low-$T$ dissipation requires rather small values of $R_m$, while $R(T)$ is rather large at high $T$. This requires the use of high values of $R_s$. Although this might seem at odds with the idea that the superconducting regions correspond to those regions where the electron density is higher, this choice of parameters can find a rationale by carefully considering the two families of carriers appearing in these LAO/STO interfaces. As discussed in Ref. [51], the 2DEG can be effectively described in terms of low-mobility and high-mobility carriers (LMC and HMC, respectively), the latter being ultimately responsible for the superconductivity onset. Indeed, one could argue that the density of states (DOS) of the superconducting regions, i.e., the effective electron mass, is large in spite of a small fraction of HMC and leads to a large local resistivity.

The mobility of these few carriers can be high if the small scattering compensates for the larger mass. At the same time, the metallic regions could have a small DOS, preventing them from becoming superconducting, but a large number of LMC can result in a comparatively smaller resistivity. To be more quantitative, the values extrapolated in [51] for the density of the two carriers $n_1, n_2$, for LMC and HMC, respectively, and their mobility $\mu_1, \mu_2$, give us the order of magnitude for the ratio $R_m/R_s$ in the UD and OD regime. In particular, $\frac{R_m}{R_s} = \frac{n_1\mu_1}{n_2\mu_2} \sim 30$ for $V_G \simeq 50\,V$ and $\frac{R_m}{R_s} = \frac{n_1\mu_1}{n_2\mu_2} \sim 100$ for $V_G \leq 25\,V$ Finally, having an estimate of the effective masses of the two carriers, we can also extrapolate the order of magnitude of the ratio $L_2/L_1 = L_f/L_0$. Being $m_2^*/m_1^* \sim 0.07$ [52] and $n_1/n_2 \sim 100$, we have $\frac{L_2}{L_1} = \frac{L_f}{L_0} \simeq \frac{m_2^*}{m_1^*}\frac{n_1}{n_2} \simeq 7$.

We report in Fig. 7 the real $\sigma_1$ and imaginary $\sigma_2$ parts of the conductivity and the DC resistivity $R$ at various $R_m$ and $L_s$ for both underdoped and overdoped regimes. We use as a reference the cases discussed in Section 5, reporting panels a and d in Fig. 3 in panels a and d of Fig. 7, hence referring to Fig. 3c-d and d-e for the corresponding probability distributions $P(T_c)$ and the fractal

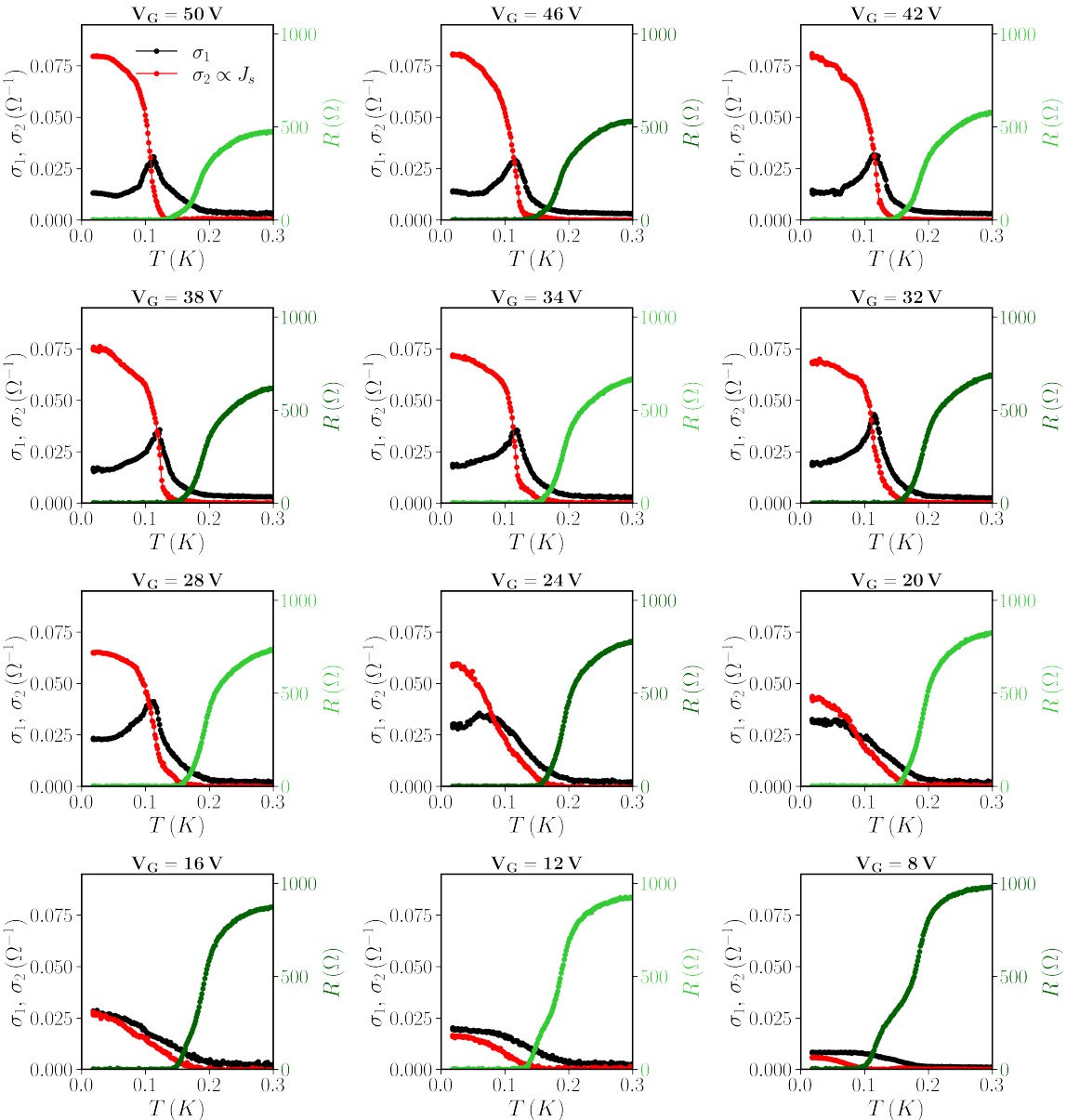

Figure 6: DC resistivity (green, right axis), real (black, left axis) and imaginary part of the conductance (red, left axis) plotted as a function of temperature for different values of the gate potential $8\,\text{V} < V_G < 50\,\text{V}$.

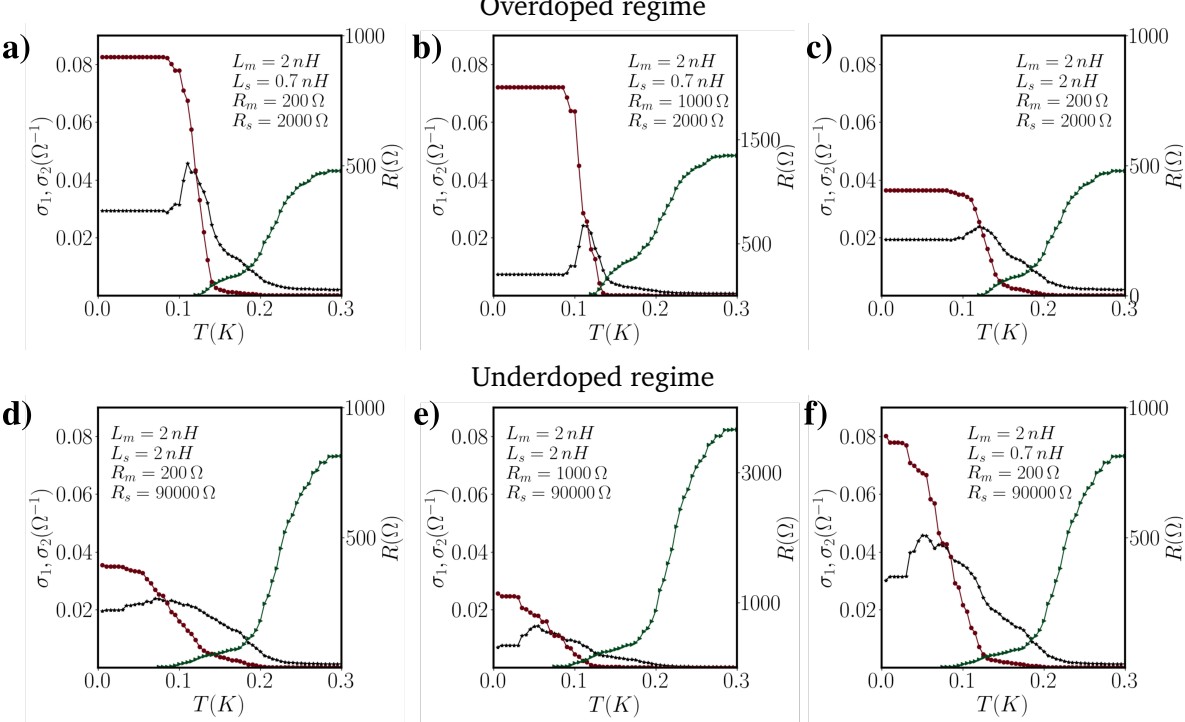

Figure 7: Temperature dependence of complex conductivity (real part in black, imaginary part in red and DC resistivity in green) and DC resistivity calculated with the RIN model using the same parameters and probability distributions of Fig. 3 (see panels c and for the OD case, panels f and g for UD). The tuning parameters are here $R_m$ and $L_s$, both acting on the superfluid response, keeping $L_m = 2$nH. OD regime: $R_s = 200\,\Omega$. (a) $L_s = 0.7$ nH and $R_m = 200\,\Omega$, (b) $L_s = 0.7$ nH and $R_m = 1000\,\Omega$, (c) $L_s = 2$ nH and $R_m = 200\,\Omega$. UD regime: $R_s = 90000\,\Omega$. (d) $L_s = 2$ nH and $R_m = 200\,\Omega$, (e) $L_s = 2$ nH and $R_m = 1000\,\Omega$, (f) $L_s = 0.7$ nH and $R_m = 200\,\Omega$.

geometry. As one can see looking at panels b and e of Fig 7, an increase of $R_m$ acts differently on both $\sigma_1$, $\sigma_2$ curves. Whereas $\sigma_2$ is only suppressed by less than $0.01\,\Omega^{-1}$ at $T = 0$, the real part of the conductivity, $\sigma_1$, gets significantly reduced by larger values of $R_m$. Conversely, a decrease in $L_s$ from a value $2$ nH to $0.7$ nH results in the increase of both $\sigma_1$ and $\sigma_2$, acting primarily on the latter one. For the OD case, this effect can be observed by comparing panels c and a, whereas for the UD case, one can compare panels d and f of Fig. 7.

It is worth noting that even if $R_m$ and $L_s$ act on the behaviours in temperature of $\sigma_1$ and $\sigma_2$, they are not enough to capture the anomalous superfluid response experimentally observed. Together with the fractal geometry, indeed, also a slight difference in the probability distributions $P(T_c)$ – namely $\sigma_f = 0.02$ K for the OD case, $\sigma_f = 0.05$ K for the UD – of the filamentary component is required in order to capture the qualitative experimental behaviour of both the OD and UD regime, as already stated in Section 5.1.

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
