# Peer review of "Superfluid response of two-dimensional filamentary superconductors"

_SciPost Physics_

## Round 1 · Referee Report · Anonymous (Referee 1) · 2023-7-16

Strengths

The paper provides theoretical studies of highly inhomogenoeous superconductors and results are confronted with the experimental data.

Weaknesses

Computational method used by the authors seems to be very efficient, but its description in the present version of manuscript is rather not fully clear.

Report

The paper by G. Venditti et al addresses the important issue of inhomogeneous
superconductivity realized in the low dimensional systems - specific examples
of experimentally observed filamentary superconducting patterns are listed
in the introductory section. Evidence for filamentary inhomogeneous nature
can be manifested for instance by a broadening of the resistive transition
or precursor gap observed above the critical temperature in tunneling spectra.
The authors focus here on 2-dimensional case, analyzing the microwave transport
properties within the 'random impedance network' (RIN) scenario. In particular,
they distinguish characteristic features of the complex conductivity resulting
from the filamentary structure of superconducting regions in the under- and
over-doped limits, respectively. Ingredients of their theoretical approach are
described in Section 3. In Sections 4 and 5 the authors analyze the complex
microwave conductance over the temperature region, ranging from below to above Tc.
They consider their calcualtions, confronting them with experimental data
for SrTiO_{3}-based heterostructures. Global transition to the superconducting
state is here driven by onset of the phase coherence, therefore the authors
study in detail the superfluid stiffness that is encoded in the imaginary
part of the complex conductance. Furthermore, they also point out some
qualitative differences of the real part conductance near Tc which occur
between under- and over-doped samples (as clearly displayed in Figure 1).
The numerical results presented in Section 5 provide important information
about the role played by geometry and disorder on the resistive and
superfluid properties for the varying gate voltage. Such results
illustrate the influence of filamentary superconducting condensate.

The paper is clearly written and theoretical results are confronted
with the experimental data. This approach might be useful to other
highly inhomogeneous superconducting systems, therefore I would
recommend the paper for publication. Before any final decision,
however, I kindly ask the authors for a few (rather technical)
explanations/amendments.

(i) In Figure 2 the complex conductivity is presented for the microwave
frequency 0.36 GHz. Is this particular choice of frequency representative
for the considerations of resistive and superfluid properties ?

(ii) Would it be feasible to obtain the frequency dependent \sigma_{1}
and \sigma_{2} using RIN technique ? If so, perhaps the authors could
present some typical plot, showing the complex conductance within
the frequency region from \omega=0 to omega=2\Delta (or broader).
Such plot might be valuable and instructive for some infrared
spectroscopy measurements on highly inhomogeneous superconductors.

(iii) For the readers less familiar with 'random resistor network' (RRN)
and its RIN extension it would be very helpful to learn more about this
computational algorithm. I urge the authors to expand their section 3
including more details in order to make the paper be self-contained.

Requested changes

Points (i)-(iii) in the main report.

  • validity: good
  • significance: high
  • originality: high
  • clarity: ok
  • formatting: good
  • grammar: perfect

Author:  Giulia Venditti  on 2023-10-20  [id 4053]

(in reply to Report 1 on 2023-07-16)
Category:
answer to question

We thank the referee for their very positive report and for recommending our paper for publication
in SciPost. In the attached file we provide a point-by-point answer to all the issues raised.

Attachment:

Referee_Report_1.pdf

---

## Round 1 · Referee Report · Anonymous (Referee 2) · 2023-9-5

Report

The authors, after a short introduction devoted to the discussion of the
phenomenon of superconductivity in heterogeneous mesoscopic systems, focus on an attempt to build a model describing quasi-two-dimensional
heterostructures in which the phenomenon of superconductivity in the
interface occurs. In particular, the subject of the work is an attempt to
describe the phenomenon of superconductivity in disordered
quasi-one-dimensional filaments ( lattice) near the percolation boundary
studying that phenomenon with a kind of a two-fluid model.
In the further part of the paper, they compare the results of the
calculations and simulations with the results obtained, as it results from
the text earlier (see below) in other publications, of measurements of the
impedance of LAO/STO heterostructures. In conclusion, the authors write
that (page 12, third paragraph):
: "were able to identify the specific physical effects of […] on the
model on macroscopic transport: the resistivity and inductance of the
various regions , how rapidly the normal metal becomes superconducting by
decreasing T, due to the width of the distribution of temperatures
associated with the microscopic disorder, and so on.".
In the respectful opinion of this reviewer, this conclusion is not justified.

Starting from the logic, from the compliance of the model results for a
certain set of parameters with the measurement results, it does not yet
follow that the model correctly describes the reality. The second problem
concerns measurements. The geometry of the measuring system, not discussed at all, but probably identical to the one described in [46], in the
language of circuit theory allows only the measurement of the amplitude
and phase of the wave reflected from the DUT. The authors write that they
are limited to a very narrow frequency region near 360 MHz, which in free
space corresponds to a wavelength of about 14". This is much more than the
0.1" size of their sample. In practice, it is a resonant RLC circuit with
a piece of quasi-2d material as partially L, R, and C, frequency and temperature dependent element, and of course with many other elements which sizes, properties, geometry and linearity contribute to the quality factor and the resonance frequency of the circuit.
The VNA (measuring instrument) tests the entire measurement chain,
cables, sockets, strip lines, etc. ending with the sample and matching
electronics, and the complex impedance, real and imaginary conductivity numbers, shown in pictures are based on a behavioral model of "Device Under Test", inhomogeneous sample in a resonant microwave system, with many distributed and discrete elements, which the authors do not write about. How large are the estimated errors in determining the values \sigma_1(T, \omega) and \sigma_2?
How repeatable are the measurements for different samples? Do the conclusions are supposed to be about a phenomenon? In fact. on page 7
of the manuscript authors write "It is wort noting that the tree peculiar
features summarized above are indeed characteristic of the rather disordered sample, whereas in more homogeneous samples some of those peculiarities can be less pronounced or even absent." What is the measure of the "sample homogeneity" and how was it determined?
Lastly, Does the model predict an unobvious phenomenon that awaits experimental confirmation?

Turning to the model, it is worth noting two elements that were almost
completely omitted in the discussion by the authors: the RIN model described on page 5 is based on the linear response of the system (in the authors' words, Ohm's law and Kirchhoff’s law). The system of even weakly coupled superconductors is a non-linear system, and the phenomenon of proximity may complicate this description even more. Discussion of these elements of the physical system is definitely missing in the work.

The last remark concerns percolation phenomena. Author's narration liberally mixes up microscopic and electrodynamic description, using term "optical" for microwave, and in fact not even microwave but radio spectroscopy. The frequency of their probe as it seems, except for the lowest temperatures, is basically indistinguishable from DC probe.
The probe energy of 1.5 micro eV is much lower than the gap of their superconductor condensate, based on Tc at approx. 0.1K and much lower than kT of their experimental conditions. Does the real part of the conductivity, determined with the Drude model or alike,
above Tc matches the DC conductivity value? If not, as it may be the case in disordered systems, how important is AC hopping conductivity in that system?

Studies of disordered systems have a long history and by the early 1970s the work of Boris I. Shklovskii and Alex L. Efros was already a classic. During the
period of intensive study of high-temperature superconductors, numerous
groups dealt with these problems, in Zuerich, in Cambridge, in the former
USSR, many other places, and a lot is known already about the reasons of Tc
broadening in inhomogeneous systems, where not only carrier density but also scattering phenomena may influence (local) Tc.

Further technical notes on text, organization, illustrations and references.

Appendix B, on page 14 and 16 of the manuscript, is a word for word
self-plagiarism of the Methods Sample growth paragraph on page 7 of the
paper cited as reference [46]. Does it implies that samples and
experimental data shown in this paper are not original, and have been
done, and published, before? It is not explained clearly enough.

Next, what is the purpose of a full page picture no. 6. in Appendix C?
Not a single word is devoted to this picture. BTW, it would be
much clearer for a physicist reader to see the axes of graphs in units
that reflect the physics of the problem rather than the engineering, such
as the ratio T/Tc for temperature, E/\delta for energy, and n/nc for
concentration, instead of the only technically relevant, like voltage, for
example gate polarity. Why 4 volts step should be relevant, is the
voltage-carrier density relation linear? It may be also worth discussing
the role, or lack thereof, of the relationship between the frequency of
the AC signal used to measure the impedance and the frequency (energy) of
the energy gap associated with the formation of the condensate. Where and
if 2\delta is < 3.5 kT?

Last, but not least, the reviewer's attention was also drawn to the
selection of references, especially the percentage of self-citations.
Without taking a position in the ongoing debate on this issue, the
reviewer believes that 33% of self-citations by a leading author in the
reference list is well outside the range considered appropriate.

In conclusion I recommend against publication of submitted manuscript in
the present form.
  • validity: -
  • significance: -
  • originality: -
  • clarity: -
  • formatting: -
  • grammar: -

Author:  Giulia Venditti  on 2023-10-20  [id 4052]

(in reply to Report 2 on 2023-09-05)
Category:
answer to question
reply to objection

We provide a point-by-point answer to all the issues raised by the referee. Please, see attached pdf file.

Attachment:

Referee_Report_2.pdf

---

## Editorial Decision

resubmitted